# Tuning structural isomers of phenylenediammonium to afford efficient and stable perovskite solar cells and modules

Cheng Liu [1,2,11], Yi Yang [1,2,11], Kasparas Rakstys [3✉], Arup Mahata [4,5], Marius Franckevicius[6], Edoardo Mosconi[5], Raminta Skackauskaite[3], Bin Ding[2], Keith G. Brooks[2], Onovbaramwen Jennifer Usiobo[7], Jean-Nicolas Audinot[7], Hiroyuki Kanda[2], Simonas Driukas[6], Gabriele Kavaliauskaite[6], Vidmantas Gulbinas[6], Marc Dessimoz[2], Vytautas Getautis[3], Filippo De Angelis[4,8,9], Yong Ding [1,2✉], Songyuan Dai [1✉], Paul J. Dyson [2✉] & Mohammad Khaja Nazeeruddin [2,10✉]

Organic halide salt passivation is considered to be an essential strategy to reduce defects in state-of-the-art perovskite solar cells (PSCs). This strategy, however, suffers from the inevitable formation of in-plane favored two-dimensional (2D) perovskite layers with impaired charge transport, especially under thermal conditions, impeding photovoltaic performance and device scale-up. To overcome this limitation, we studied the energy barrier of 2D perovskite formation from *ortho*-, *meta*- and *para*-isomers of (phenylene)di(ethylammonium) iodide (PDEAI$_2$) that were designed for tailored defect passivation. Treatment with the most sterically hindered *ortho*-isomer not only prevents the formation of surficial 2D perovskite film, even at elevated temperatures, but also maximizes the passivation effect on both shallow- and deep-level defects. The ensuing PSCs achieve an efficiency of 23.9% with long-term operational stability (over 1000 h). Importantly, a record efficiency of 21.4% for the perovskite module with an active area of 26 cm$^2$ was achieved.

[1] State Key Laboratory of Alternate Electrical Power System with Renewable Energy Sources, North China Electric Power University, Beijing 102206, People's Republic of China. [2] Institute of Chemical Sciences and Engineering, EPFL VALAIS, Sion 1951, Switzerland. [3] Department of Organic Chemistry, Kaunas University of Technology, Radvilenu pl. 19, Kaunas 50254, Lithuania. [4] CompuNet, Istituto Italiano di Tecnologia, Via Morego 30, 16163 Genova, Italy. [5] Computational Laboratory for Hybrid/Organic Photovoltaics (CLHYO), Istituto CNR di Scienze e Tecnologie Chimiche "Giulio Natta" (CNR-SCITEC), Via Elce di Sotto 8, 06123 Perugia, Italy. [6] Department of Molecular Compound Physics, Center of Physical Sciences and Technology, Sauletekio av. 3, LT-10257 Vilnius, Lithuania. [7] Advanced Instrumentation for Nano-Analytics (AINA), Materials Research and Technology Department, Luxembourg Institute of Science and Technology (LIST), L-4422 Belvaux, Luxembourg. [8] Department of Chemistry, Biology and Biotechnology, University of Perugia, Via Elce di Sotto 8, 06123 Perugia, Italy. [9] Department of Mechanical Engineering, College of Engineering, Prince Mohammad Bin Fahd University, P.O. Box 1664 Al Khobar 31952, Kingdom of Saudi Arabia. [10] Department of Materials Science and Engineering, City University of Hong Kong, Kowloon, Hong Kong. [11]These authors contributed equally: Cheng Liu, Yi Yang. ✉email: kasparas.rakstys@ktu.lt; dingy@ncepu.edu.cn; sydai@ncepu.edu.cn; paul.dyson@epfl.ch; mdkhaja.nazeeruddin@epfl.ch

As the front runner among emerging photovoltaic technologies, perovskite solar cells (PSCs) with certified power conversion efficiencies (PCEs) over 25% show great promise for scale-up and future commercialization due to relatively simple and low-cost solution processes[1–4]. However, the disordered stoichiometric compositions at surfaces, the loss of organic components during thermal annealing, and the heterogeneous polycrystalline nature inevitably generate abundant defects in the solution-processed perovskite films, particularly at surfaces and grain boundaries[5–8]. Such defects incur electronic states in the bandgap of the perovskite and behave as non-radiative recombination centers, which shorten the carrier lifetime and limit the photovoltaic performance[9]. Moreover, these defects are responsible for local charge accumulation, accelerated ion migration, and the initial invasion of moisture or oxygen, ultimately causing device instability issues[3]. The defects also hinder the scale-up of PSCs to modules, thus restricting commercialization[10].

To address this problem, various defect reduction methods have been proposed including (i) composition tuning[11–13], (ii) crystal growth regulation[14–16], and (iii) surface passivation[17–20]. To date, surficial post-treatment by alkylammonium halides is commonly exploited to achieve efficient and stable perovskite devices, and many compounds have been evaluated[21–24]. Typically, an additional two-dimensional (2D) perovskite layer is formed on top of the primary perovskite absorber after treatment with alkylammonium halides, improving the stability of the devices[8,25,26]. A stubborn in-plane orientation and high exciton binding energy are usually observed for the surficial 2D perovskite layer, which potentially suppresses charge transport and draws back the passivation effect, especially when using bulkier spacer cations[27–29]. For this reason, recently, You et al. removed the annealing process after post-treatment to enable a thin phenylethylammonium iodide (PEAI) layer instead of a 2D perovskite layer for more effective surface passivation[30]. As a result, they achieved a certified PCE of over 23%, benefiting from a combination of promotion of charge transport by the π-conjugated phenyl rings, reduction of $Pb^{2+}$ interstitials through amine coordination, and filling of iodine vacancies by iodide ions[17]. However, these devices still suffered from a plummet in PCE under higher operating temperatures due to the conversion of PEAI to 2D $PEA_2PbI_4$. Therefore, an alkylammonium halide that can sustain higher temperatures without undergoing 2D perovskite formation is highly desirable to effectively passivate surface defects for efficient and stable PSCs and modules.

Herein, we describe surface passivation of perovskite films using ortho-(phenylene)di(ethylammonium) iodide (o-PDEAI$_2$) that leads to high-performance PSCs and modules. From an investigation of the ortho-, meta-, and para-isomers for PDEAI$_2$, the ortho-isomer effectively increases the energy barrier of the 2D perovskite formation and prevents the bulky organic cations from entering the perovskite lattice even at elevated temperatures. Surficial o-PDEAI$_2$ exhibits a comprehensive passivation effect on both shallow- and deep-level defects, which suppresses nonradiative recombination and improves interfacial charge extraction, resulting in a PCE of 23.9% in PSCs with enhanced long-term stability at ambient and elevated temperatures, and light-soaking conditions. Furthermore, a o-PDEAI$_2$-based perovskite module, with an active area of 26 cm$^2$, presents a record efficiency of 21.4%.

## Results

### Isomer construction for suppressing 2D perovskite formation.
The ortho-, meta-, and para-isomers of PDEAI$_2$ (Fig. 1a) were prepared and further details are provided in the Supplementary

Information. To examine their potential to form 2D perovskites, films composed of equimolar quantities of PDEAI$_2$ and PbI$_2$ were fabricated. The X-ray diffraction (XRD) patterns of the para-PDEAI$_2$ (p-PDEAI$_2$)-based film (Fig. 1b) exhibit a dominant peak at ~7.15° associated with (002) lattice reflections of the 2D (p-PDEA)PbI$_4$ perovskite[31,32]. The meta-PDEAI$_2$ (m-PDEAI$_2$)-based film contains the same peak, but with a much lower relative intensity, implying that the 2D perovskite is formed, but to a much lower extent. The non-perovskite amorphous XRD pattern observed in the o-PDEAI$_2$-based film suggests that the 2D perovskite cannot be formed. Absorption and photoluminescence (PL) spectra reveal a similar trend (Fig. 1c). The p-PDEAI$_2$-based film shows an exciton absorption around 516 nm and a PL emission around 532 nm, assigned to the characteristic peaks of 2D perovskite with layer thickness $n = 1$[33]. These optical characteristics were not observed in the m-PDEAI$_2$-based sample due to the negligible 2D component in the film. Similarly, no absorption or emission signal was identified in the region 500–600 nm for o-PDEAI$_2$-based film, indicating that the formation of 2D perovskite is disfavored.

Next, PDEAI$_2$ isopropanol solutions were spin-coated onto the perovskite surface and experiments were performed to determine whether the surficial 2D perovskite is formed during the post-treatment process, especially under thermal induction. Widely used PEAI was employed for comparison. Figure 1d displays the temperature-dependent PL spectra of perovskite films under 450 nm excitation. As the temperature increases, the films deposited with PEAI and p-PDEAI$_2$ show enhanced emission in the 510–540 nm region, evidencing the growth of 2D perovskite with $n = 1$ on the film surface (Fig. 1b). In contrast, post-treatment with m-PDEAI$_2$ and o-PDEAI$_2$ did not lead to the formation of surficial 2D perovskite, even at elevated temperatures. This was further confirmed by the temperature-dependent XRD (Fig. 1e and Supplementary Fig. 2), where reinforced (002) reflections of 2D perovskite were observed for the p-PDEAI$_2$- and PEAI-treated films whereas no peak was observed for the other two films. The upper 2D (p-PDEA)PbI$_4$ perovskite shows a lower diffraction intensity than that of the (PEA)$_2$PbI$_4$, which is commonly observed for other Dion-Jacobson (DJ) perovskites[31,34]. Grazing-incidence wide-angle X-ray scattering (GIWAXS) was used to probe the crystal orientation of the annealed films (Fig. 1f). The surficial (PEA)$_2$PbI$_4$ exhibits a combined feature of diffraction arcs and Bragg spots along the $q_z$ direction at 0.39 Å$^{-1}$ and 0.78 Å$^{-1}$, implying a random but predominantly in-plane orientation of the (002) and (004) crystal planes. A similar orientation was observed for (p-PDEA)PbI$_4$ with the characteristic (002) crystal plane at $q_z = 0.51$ Å$^{-1}$. Only diffraction signals corresponding to the three-dimensional (3D) perovskite ($q_z = 1.00$ Å$^{-1}$) and excess PbI$_2$ ($q_z = 0.91$ Å$^{-1}$) were identified for the m-PDEAI$_2$- and o-PDEAI$_2$-treated films, which is consistent with the XRD results (Supplementary Fig. 2). It is evident that both the monoammonium and the para-diammonium substituents on the phenyl ring enable organic halide salts to react with PbI$_2$ from the underlying 3D perovskite during the post-treatment process and form an in-plane-orientation dominated 2D perovskite on the surface (Fig. 1g). The arbitrary, especially in-plane, orientation will block the charge transport along with the inorganic framework due to the alternant insulating organic layers[35]. In contrast, m-PDEAI$_2$ and o-PDEAI$_2$ form a thin organic halide salt layer instead of 2D perovskite layer on the surface, which may avoid the energy disorder and the efficiency loss at elevated operating temperatures.

Density functional theory (DFT) calculations were used to understand the specific behavior of each PDEAI$_2$ isomer. The formation energies of 2D perovskites formed from p-PDEAI$_2$,

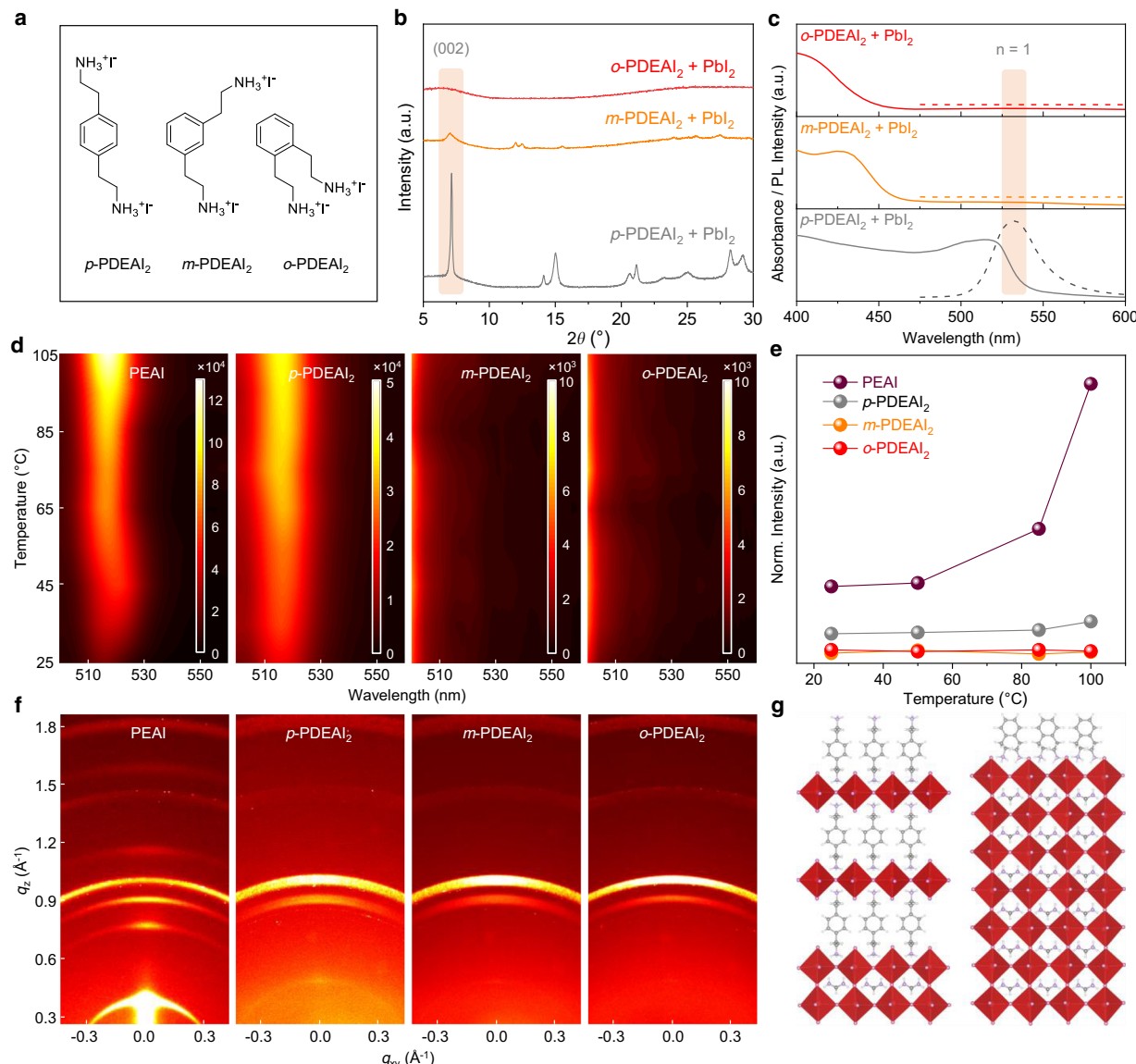

**Fig. 1 Structures of PDEAI₂ and subsequent formation of 2D perovskites. a** The structures of the PDEAI₂ isomers. **b** XRD patterns of the films obtained from equimolar ratios of PDEAI₂ and PbI₂. **c** Absorption and PL emission spectra of the films obtained from equimolar ratios of PDEAI₂ and PbI₂. **d** Temperature-dependent PL images of the perovskite films treated with the different organic halide salts. **e** Intensity of the (002) reflections of the surficial 2D perovskites as a function of annealing temperature for the post-treated perovskite films. **f** GIWAXS images of the perovskite films following post-treatment. **g** Schematic illustration of $p$-PDEAI₂ and $o$-PDEAI₂-assembled perovskite surface.

$m$-PDEAI₂ or $o$-PDEAI₂ were calculated (Fig. 2a and Supplementary Table 1). The formation of 2D perovskite with $p$-PDEAI₂ and $m$-PDEAI₂ are energetically more favorable than $o$-PDEAI₂. For comparison, the formation energy of 2D perovskite formed from PEAI was also calculated to be −7.71 eV (Supplementary Fig. 3), which is lower than that of PDEAI₂ isomers and consistent with experimental observation (Fig. 1d, e and Supplementary Fig. 2). We further checked the possibility of the formation 2D perovskite with a larger layer thickness and calculated the reaction thermodynamics of the process with the different isomers. Supplementary Fig. 4 and Supplementary Table 1 indicate that the reactions of $p$-PDEAI₂ and $m$-PDEAI₂ processes are more favorable than that of $o$-PDEAI₂. This difference is due to the suitable matching between the distance of two −CH₂−CH₂−NH₃⁺ groups of the cations and the distance of adjacent octahedral voids on the perovskite. As can be seen in Supplementary Fig. 5, the lowest mismatching distances are 0.85,

0.15, and 1.85 Å for $p$-PDEAI₂, $m$-PDEAI₂, and $o$-PDEAI₂, respectively. The formation of quasi-2D perovskite is also likely to be energetically costly for the $o$-PDEAI₂, which is nicely reflected by the reaction energy values. Thus, the model demonstrates that $o$-PDEAI₂ possesses the highest formation energy barrier of surficial 2D perovskite among the PDEAI₂ isomers, in good agreement with the experimental observations.

**Passivation effects on photovoltaic performance.** In order to investigate the surface passivation effect, adsorption energies were calculated by depositing the PEA, $p$-PDEA, $m$-PDEA, and $o$-PDEA cations on the perovskite surface considering both the ethylammonium and phenyl ring adsorption modes (Fig. 2b and Supplementary Fig. 6, Supplementary Table 2). The calculations indicate that all the cations show similar surface passivation if only adsorption of one molecule is considered. However,

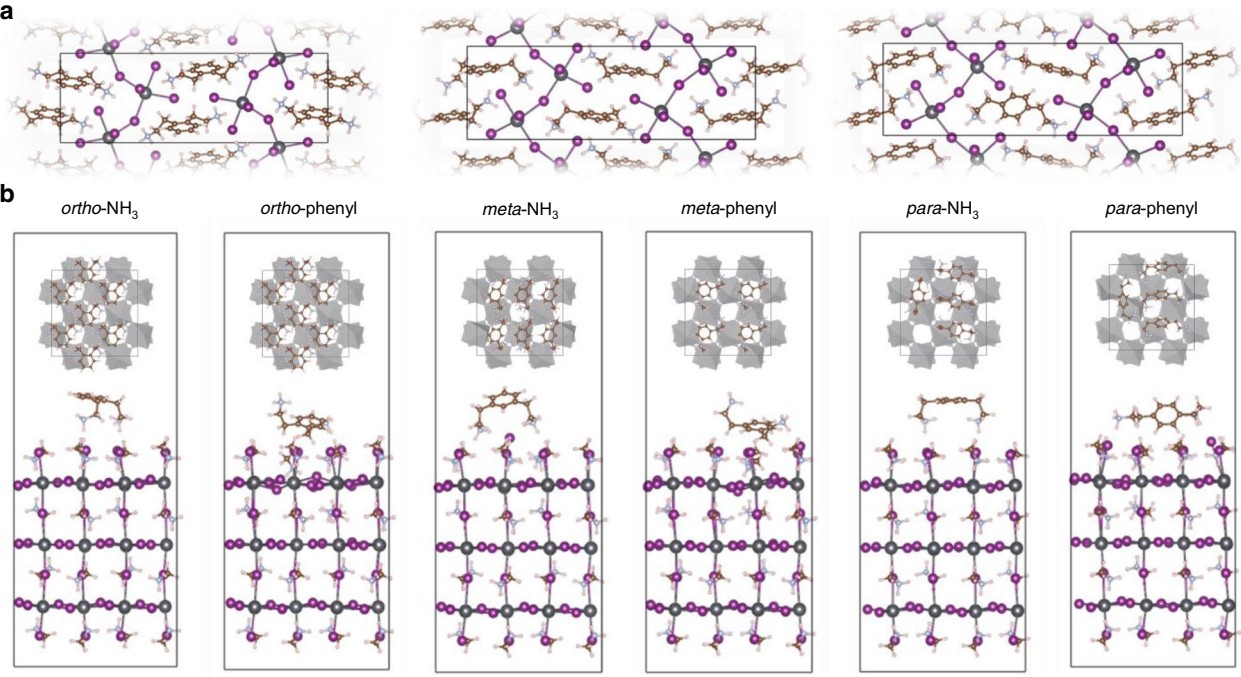

**Fig. 2 Theoretical modeling. a** Optimized 2D perovskites structures incorporating $o$-PDEAI$_2$, $m$-PDEAI$_2$, and $p$-PDEAI$_2$. **b** Optimized structures for the adsorption of 2D cations on the perovskite surface. The top insets indicate the maximum number of PDEA cations that can be adsorbed on a 2 × 2 perovskite surface. Schematic is based on the optimized coordinates of the corresponding structures and translating them in multiple numbers. For the $o$-PDEAI$_2$ structures, both the modes adsorb in a similar manner.

passivation is achieved by the cumulative effect related to the number of cations that can be adsorbed per unit surface area. The optimized structures were processed to estimate the maximum number of cations that may be adsorbed on a 2 × 2 supercell (17.7 Å$^2$) of perovskite surface, as illustrated in the insets of Fig. 2b. For the $p$-PDEA cation, only 4 cations can be adsorbed, increasing to 4–6 for $m$-PDEA and 6 for $o$-PDEA cations. Therefore, it can be qualitatively argued that the $o$-PDEA cation shows a better surface passivation effect due to the increased binding affinity and higher coverage, which may reduce the possibility of defect formation.

To examine the passivation effect, we incorporated the PDEAI$_2$ isomers onto the perovskite film with a surficial morphology corresponding to that shown in Supplementary Fig. 7, and the photovoltaic devices were fabricated with a typical configuration consisting of fluorine-doped tin oxide (FTO)/TiO$_2$/SnO$_2$/perovskite/PDEAI$_2$/2,2′,7,7′-tetrakis-($N$,$N$-di-4-methoxyphenylamino)-9,9′spirobifluorene (spiro-OMeTAD)/Au (Fig. 3a). Current density ($J$)–voltage ($V$) characteristics of the devices are compared in Supplementary Fig. 8a and Supplementary Table 3. As expected, the devices with $p$-PDEAI$_2$ show slightly lower PCEs compared to the control devices, ascribed to the high binding energy and electrical anisotropy of 2D ($p$-PDEA)PbI$_4$ layer that hinders the charge separation and out-of-plane charge transport[29]. Interestingly, a comparatively low efficiency was observed for the $m$-PDEAI$_2$-based devices. This may be rationalized by the two –CH$_2$–CH$_2$–NH$_3^+$ groups of $m$-PDEA anchoring at adjacent octahedral voids of the perovskite surface (Supplementary Fig. 5), which might cause carrier stagnation and resultant charge recombination. Among the three isomers, $o$-PDEAI$_2$ has the best passivation effect, improving the average PCE from 20.88% of the control device to 22.41%, mainly reflected by the enhanced fill factor (FF) and open-circuit voltage ($V_{OC}$). The effective passivation of $o$-PDEAI$_2$ may be attributed to (i) the high adsorption density on the perovskite, (ii) the strong

adsorption energy due to the diammonium structure, and (iii) the electrostatic interactions with the defective perovskite surface[17,36]. As a result, the best-performing PSC passivated with $o$-PDEAI$_2$ shows a PCE of 23.92%, with a $V_{OC}$ of 1.157 V, a short-circuit current ($J_{SC}$) of 24.75 mA cm$^{-2}$, and an FF of 83.50%, in which the concentration of the $o$-PDEAI$_2$ was optimized to be 2 mg mL$^{-1}$ (Supplementary Table 4), compared to the best control device with a PCE of 21.94%, $V_{OC}$ of 1.135 V, $J_{SC}$ of 24.49 mA cm$^{-2}$ and FF of 79.00%. The integrated $J_{SC}$ values from the external quantum efficiency (EQE) spectra in Supplementary Fig. 8b matches well with those measured from the $J$–$V$ curves (< 1% discrepancy). A stabilized power output of 23.50% and negligible hysteresis are demonstrated for the $o$-PDEAI$_2$-based device as shown in Fig. 3b. Also, the statistic histogram reveals good reproductivity with 75% of the devices having PCEs exceeding 22% (Fig. 3c), confirming the effectiveness of $o$-PDEAI$_2$ passivation.

To elucidate the passivation effect on the hole extraction and carrier dynamics in the devices, voltage-dependent transient PL and transient photocurrent studies were performed. Supplementary Fig. 9a–e shows the PL decay kinetics obtained for different devices at various applied voltages. All devices show bicomponential PL decay, and the dependences of the average decay times on the applied voltage are shown in Fig. 3d. The applied voltage reduces the average decay times, indicating that they are at least partly determined by the carrier extraction. The average decay times anticorrelate with the device efficiency and particularly with FFs, where the devices with faster PL decays show better performance. Specifically, $o$-PDEAI$_2$ passivation causes the fastest PL decay attributed to the improved hole extraction, whereas the $m$-PDEAI$_2$ passivated film shows much slower PL decay, which indicates impaired hole extraction deteriorating the device performance[37].

To exclude the influence of the built-in electric field present in solar cells, we have additionally investigated the PL decays in the

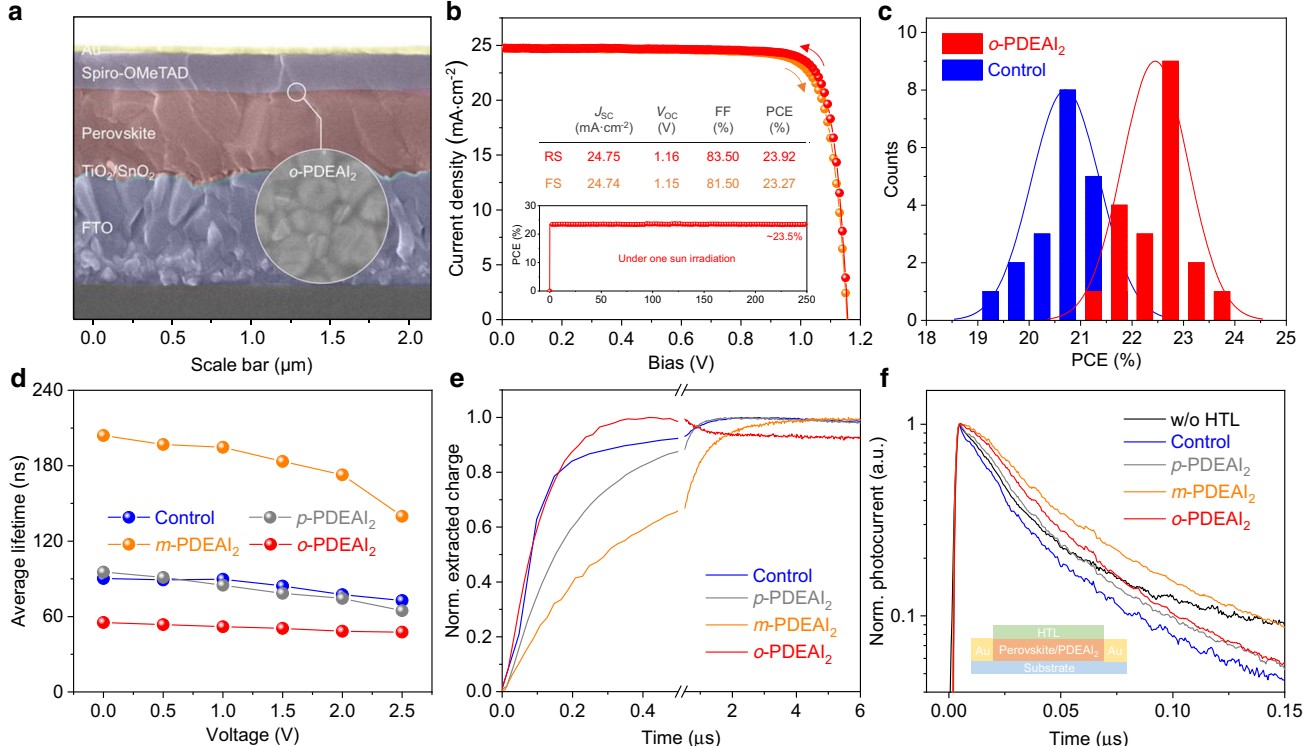

**Fig. 3 Passivation layer-dependent device performance. a** The cross-sectional scanning electron microscope (SEM) image of the PSC passivated with $o$-PDEAI$_2$. **b** $J$–$V$ characteristics of the champion device with $o$-PDEAI$_2$ measured in both reverse (red) and forward (orange) scanning directions (the inset shows its stabilized power output). **c** The statistics of PCE distribution for devices with and without $o$-PDEAI$_2$. (Statistics from 20 samples for each condition). The solid lines denote the Gauss distribution fitting for the PCE statistics. **d** Voltage-dependent average PL lifetimes of the different devices. **e** Integrated charge extraction curves of devices. **f** Transient photocurrent decay in lateral sample geometry.

treated with PDEAI$_2$ and nontreated perovskite films covered with spiro-OMeTAD deposited on glass substrates (see Supplementary Fig. 9f for kinetics and Supplementary Table 5 for bi-exponential fitting parameters). For comparison, we have also investigated a pure nontreated perovskite film, which showed a long lifetime of 284 ns indicating a high material quality[38]. The spiro-OMeTAD layer significantly shortens the PL decay due to efficient hole extraction. Passivation with the different PDEAI$_2$ isomers shows qualitatively similar results as for the complete devices confirming their influence on the hole extraction rate.

The carrier extraction dynamics were revealed by transient photocurrent measurements. Due to the high capacitance of the devices restricting the time resolution of photocurrent measurements, the carrier extraction investigations were performed in integral mode measuring discharging of the device capacitance by photocurrents, when voltage was applied through a high (1 kΩ) resistor. The signal growth, in this case, corresponds to the cumulated extracted charge. Figure 3e shows the normalized charge extraction kinetics under an applied voltage of 1 V. The $o$-PDEAI$_2$-passivated device shows the fastest carrier extraction of <0.2 µs. (The partial signal drop during about 0.4–2.0 µs is caused by the sample recharging by the capacitance from the external circuit.) The control device shows similarly fast initial extraction, however, an additional slow extraction component is also observed. Significantly slower charge extraction from the $m$-PDEAI$_2$-treated device is in good agreement with the PL decay data.

We further addressed the role of PDEAI$_2$ passivation by investigating transient photocurrent kinetics in lateral sample geometry. The perovskite films were formed on an interdigitated comb of Pt electrodes (IDE) with 5 µm interelectrode distances and a spiro-MeOTAD layer was deposited on top of perovskite

films. Due to the large, 5 µm interelectrode distance, the carrier extraction was slow. We also used low excitation intensity, when carrier recombination could be ignored. In these conditions, the photocurrent kinetics on a sub-microsecond time scale was mainly determined by the carrier trapping and by the hole extraction to hole transport layer (HTL)[39]. The control device with HTL shows faster photocurrent decay than that without HTL during initial ~50 ns due to the hole extraction (Fig. 3f). The influence of PDEAI$_2$ treatment is expected to be bilateral: passivation of surface traps shall cause slower photocurrent decay, and modification of the hole extraction rate may cause additional photocurrent changes. Indeed, all passivated samples show slower photocurrent decay indicating reduced carrier trapping. The $m$-PDEAI$_2$-treated sample shows particularly slow decay, apparently because of reduced carrier trapping and the prevented hole extraction. While the $o$-PDEAI$_2$-treated sample shows similar slow carrier decay during initial ~20 ns, due to reduced carrier trapping, but at longer times the photocurrent decays faster attributed to the improved hole extraction. The overall transient PL and photocurrent results provide indications of interfacial trap passivation by the PDEAI$_2$ isomers and also demonstrate that $o$-PDEAI$_2$ treatment reduces the carrier transfer barrier at the perovskite/HTL junction and effectively increases the hole extraction rate contributing to the FF enhancement of PSCs.

Ultraviolet photoelectron spectroscopy (UPS) was used to understand the surface energy band structure with and without $o$-PDEAI$_2$. The work function is determined to be −4.45 eV and −4.91 eV for the pristine perovskite surface and the $o$-PDEAI$_2$ passivated surface, respectively (Supplementary Fig. 10). This shift is in accordance with the change in surface potential probed by Kelvin probe force microscopy (KPFM) as reflected in Fig. 4a.

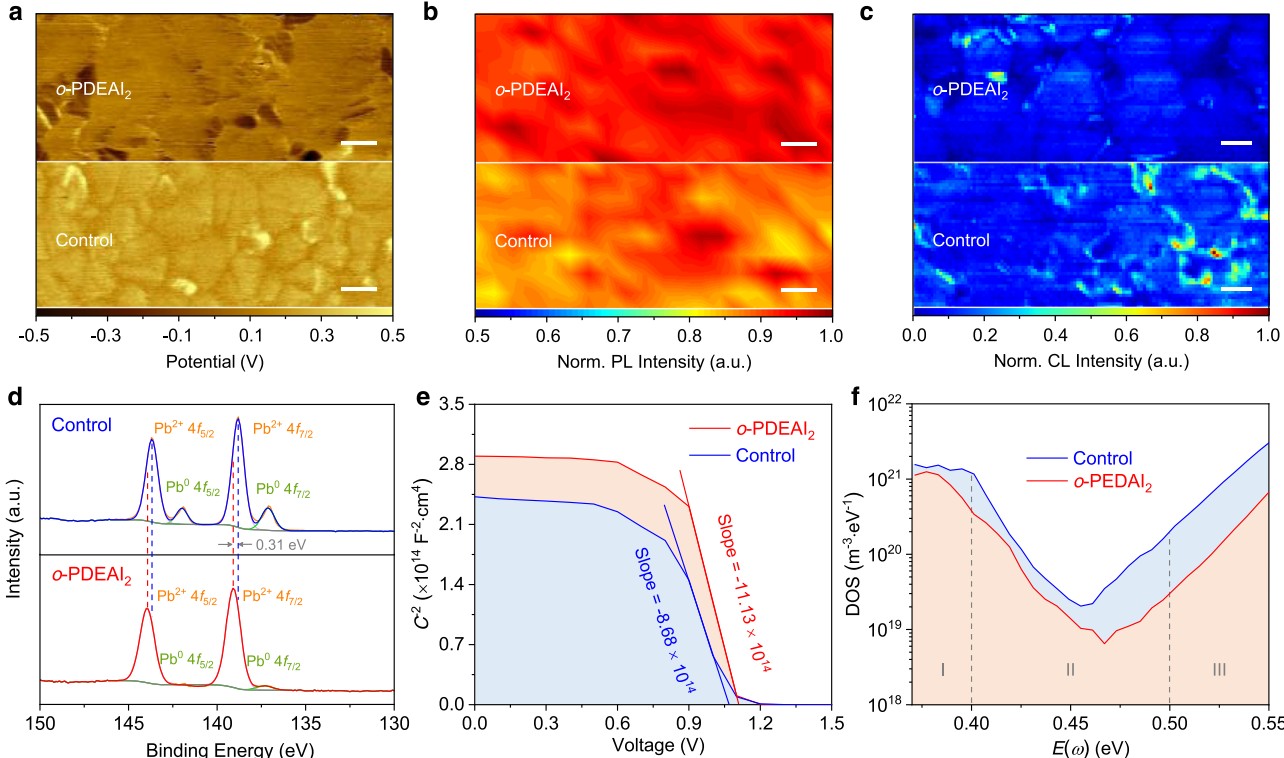

**Fig. 4 Characterization of the perovskite films and interface with *o*-PDEAI₂. a** KPFM images of perovskite films showing the electronic chemical potential. **b** Normalized PL intensity maps of the perovskite films illustrating the film homogeneity. **c** Normalized CL mapping images of the perovskite films. **d** High-resolution Pb 4*f* core-level XPS spectra of perovskite films. **e** Mott–Schottky plots of the devices revealing the interfacial charge density. **f** tDOS distribution in PSCs showing less trap densities after *o*-PDEAI₂ passivation.

The *o*-PDEAI₂-passivated surface exhibits a lower average electronic chemical potential (0.16 mV) than that of the control (0.33 mV). The downshifted energy level suggests a less *n*-type surface after *o*-PDEAI₂ treatment, which causes upwards band bending and improves hole transport and extraction for higher $V_{OC}$ values[40,41]. Confocal PL microscopy mapping was applied to probe the homogeneity of perovskite films. The film with *o*-PDEAI₂ presents more uniform PL emission compared to the control film, indicating less surface defect densities (Fig. 4b). This is also revealed by the cathodoluminescence (CL) intensity maps, where a more homogeneous distribution was demonstrated for the film with *o*-PDEAI₂ passivation (Fig. 4c). To complement these data the corresponding CL spectra (Supplementary Fig. 11) confirm that *o*-PDEAI₂ treatment does not induce 2D perovskite formation on the surface and the CL peaks at around 508 nm may be assigned to the presence of excess PbI₂ in both films. To confirm the presence of *o*-PDEAI₂ on the perovskite surface, X-ray photoelectron spectroscopy (XPS) was performed. The high-resolution XPS patterns of C 1*s* (Supplementary Fig. 12) show an emerging peak at ~291.0 eV, corresponding to the conjugated π–π bonding of the phenyl group in *o*-PDEAI₂[30]. The XPS pattern of Pb 4*f* for the control film displays two dominant peaks located at 138.8 and 143.7 eV, assigned to the Pb 4$f_{7/2}$ and Pb 4$f_{5/2}$, respectively (Fig. 4d), which shift to 139.1 and 144.0 eV for the film with *o*-PDEAI₂. This shift to higher binding energies reflects the modified electronic states of the perovskite atoms due to interactions between *o*-PDEAI₂ and the perovskite surface[41]. In addition, another two peaks located at 137.0 and 141.9 eV arising from the metallic Pb were observed in the control film, which were not observed in the passivated film. This implies that *o*-PDEAI₂ can combine with under-coordinated Pb²⁺ ions and thereby avoid the formation of metallic Pb, thus restraining the

ionic defects and improving the operational stability of the PSCs[42]. To quantify the passivation effect of *o*-PDEAI₂, a Mott–Schottky analysis and trap density of states (tDOS) measurements were conducted on the complete devices. As indicated in Fig. 4e, the higher built-in potential (1.146 vs 1.121 V) and lower interfacial charge density (2.73 × 10¹⁵ vs 3.50 × 10¹⁵ cm⁻³) for the *o*-PDEAI₂-passivated device demonstrates that the *o*-PDEAI₂ layer can enhance the driving force of carrier injection and reduce the recombination losses at the interface[10]. The tDOS distribution under each energy demarcation was characterized as shown in Fig. 4f. The charging and discharging of trap states within the bandgap contributes to a larger variation in capacitance with the change of frequencies. Thus, the frequency differential capacitance can disclose the defects at certain energy demarcations[43]. It was found that the control device has higher trap densities of 10¹⁹–10²¹ m⁻³ eV⁻¹ over three trap bands. After *o*-PDEAI₂ treatment, the tDOS was reduced over the entire energy region, to 10¹⁸–10²⁰ m⁻³ eV⁻¹, confirming that the *o*-PDEAI₂ effectively passivate the defects with both shallow and deep energy levels on the surface of perovskite film.

We further evaluated the scalability of surface passivation by *o*-PDEAI₂ by fabricating a perovskite solar module with a total active area of 26.00 cm² (Fig. 5a). The module was completed by laser etching with 9 subcells in series and the interconnection is schematically illustrated in Fig. 5b. Encouragingly, the perovskite solar module based on *o*-PDEAI₂ passivation exhibits a $V_{OC}$ of 10.30 V, a $J_{SC}$ of 2.71 mA cm⁻², an FF of 76.40%, and a PCE of 21.36% (Fig. 5c and Supplementary Fig. 13), which corresponds to a designated-area efficiency of 19.27%. This is one of the highest efficiencies reported so far for perovskite solar modules (Supplementary Table 6)[44]. The high module performance

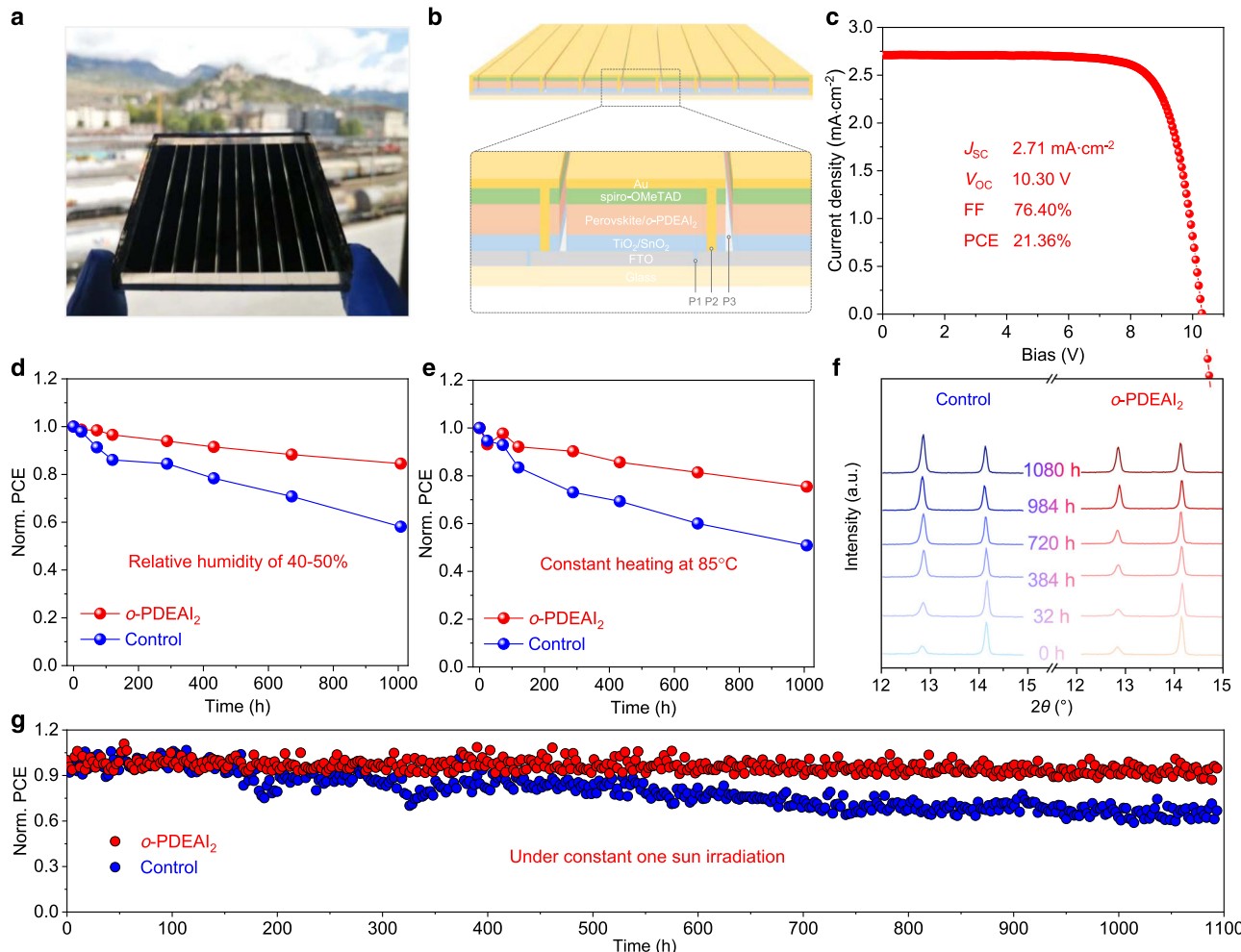

**Fig. 5 Module performance and long-term stability of the PSCs. a** Photograph of the fabricated perovskite solar module. **b** Schematic showing the interconnections of the module. **c** J–V characteristics of the champion perovskite solar module with an active area of 26.00 cm$^2$. **d** Evolution of the PCEs for the unencapsulated devices exposed to a controlled relative humidity (RH) of 40–50% in the dark. **e** Evolution of the PCEs for the unencapsulated devices under constant heating at 85 °C in a nitrogen atmosphere. **f** XRD patterns of the perovskite films as a function of annealing time at constant 85 °C. **g** Maximum power point tracking (MPPT) for 1100 h of the unencapsulated devices under continuous light (100 mW cm$^{-2}$) illumination at 25 °C in a nitrogen atmosphere.

originates from the good uniformity of the perovskite layer, the reduced trap density, and suppressed interfacial recombination, confirming the utility of the *o*-PDEAI$_2$ passivation strategy for scale-up of PSCs.

**Shelf life and operational stability**. Long-term stability tests were carried out to study the influence of *o*-PDEAI$_2$ layer on device stability. The PCEs of the unencapsulated devices under a RH of 40–50% were first tracked over time (Fig. 5d). The control device maintains 58% of the initial PCE (20.82%) after 1008 h, compared to that of 85% for the *o*-PDEAI$_2$ passivated device with an initial efficiency of 22.65%. The thermal stability was also evaluated by heating the perovskite films and corresponding devices at 85 °C in a nitrogen atmosphere. As shown in Fig. 5e, the *o*-PDEAI$_2$ passivated device (initial PCE 22.38%) degrades by 25% over 1000 h, compared to a 49% decrease for the control device (initial PCE 20.32%). Interestingly, an efficiency plummet is not observed at the start of the annealing process. That is, the suppression of converting *o*-PDEAI$_2$ into 2D perovskite effectively retains the passivation effect under constant heating, and avoids the PCE drop at the initial stage, which was normally observed in the widely used PEAI passivation[30]. XRD tracking shows that the

thermal degradation of these perovskite films results from the decomposition of the perovskite material (Fig. 5f). The control film shows an increase in the intensity of the peaks that correspond to PbI$_2$, whereas decomposition is retarded after *o*-PDEAI$_2$ passivation, indicating the *o*-PDEAI$_2$ passivation layer reduces the release of volatile organic components and enhances the resistance to heat of the perovskite material[45]. The continuous performance of the unencapsulated PSCs was also examined by MPPT under 1 sun illumination in an inert atmosphere (Fig. 5g). The device passivated with *o*-PDEAI$_2$ exhibits enhanced light stability, maintaining >90% of its original PCE after 1100 h, outperforming the control device, which degrades to >60% of the initial PCE. The robust ambient and operational stability of the *o*-PDEAI$_2$-based device may be attributed to the hydrophobicity of the phenyl group, the mitigated interfacial charge accumulation, and the suppressed ion migration benefiting from the passivation of defects[46,47].

## Discussion

We demonstrate that *o*-PDEAI$_2$ is an effective passivation agent which affords highly efficient and stable PSCs. Experimental and theoretical studies indicate that locating the two ammonium

cations in the most sterically hindered *ortho* position endows *o*-PDEAI$_2$ with an energy barrier for the formation of the in-plane favored 2D perovskite together with stronger defect passivation effects compared to the other structural isomers, to significantly reduce interfacial charge recombination. Passivation with *o*-PDEAI$_2$ boosts the efficiency of PSCs and modules (active area of 26 cm$^2$) to 23.9% and 21.4%, respectively, while ensuring long-term stabilities over 1000 h. This study demonstrates that altering functional groups and chemical structures should be an effective strategy to develop novel organic cation passivators with continuous and stable passivation effect, which may pave the way for scaling up perovskite photovoltaics to sizes of commercial relevance.

## Methods

**Device fabrication**. The TiO$_2$ compact (c-TiO$_2$) layer was deposited onto the cleaned FTO (Nippon Sheet Glass, TEC8) substrates by spray pyrolysis of titanium diisopropoxide bis(acetylacetonate) (75 wt.% in isopropanol, Sigma-Aldrich) diluted in anhydrous isopropanol (99.8%, Acros Organics) with 1:20 volume ratio at 450 °C and then annealed at 450 °C for 30 min. TiO$_2$ paste (30NR-D, GreatCell Solar) was diluted in anhydrous ethanol (99.8%, Acros Organics) with a weight ratio of 1:10 and spin-coated onto substrates at 3000 rpm for 20 s. The films were then annealed at 500 °C for 30 min in air. After cooling, 0.1 M SnCl$_4$ (99%, Acros Organics) aqueous solution was deposited on the substrate at 3000 rpm for 30 s. The films were then dried at 190 °C for 1 h and transferred into a nitrogen glovebox. 1.45 M perovskite precursors (PbI$_2$:PbBr$_2$:FAI:MAI:CsI:MACl = 0.98:0.02:0.81:0.04:0.05:0.20) were dissolved in the mixed DMF (99.8%, Acros Organics): DMSO (99.7%, Acros Organics) = 4:1 (volume ratio) solvent. The perovskite solution was spin-coated on substrates at 1000 rpm for 12 s and 5000 rpm for 28 s, respectively. 200 μL of chlorobenzene (99.8%, Acros Organics) was dropped onto the spinning films at 15 s before the end of the second step, followed by annealing at 100 °C for 10 min and 150 °C for 10 min. For the post treatment, phenyldiethylammonium iodide (PDEAI$_2$) solutions in anhydrous isopropanol (99.8%, Acros Organics) were deposited on the perovskite films at 4000 rpm for 20 s, followed by annealing at 100 °C for 5 min. 0.06 M spiro-OMeTAD was dissolved in chlorobenzene (99.8%, Acros Organics), adding 0.2 M 4-*tert*-Butylpyridine (98%, Sigma-Aldrich), 0.03 M bis(trifluoromethanesulfonyl)-imide lithium salt (LiTFSI, 99.0%, Sigma-Aldrich) in acetonitrile (99.9%, Acros Organics) and 0.0035 M FK209 Co(III) TFSI salt (GreatCell Solar) in acetonitrile (99.9%, Acros Organics). The spiro-OMeTAD solution was spin-coated at 3000 rpm for 20 s, followed by thermal evaporation of a 70 nm gold electrode under a high vacuum. For module fabrication, 6.5 cm × 7 cm FTO substrates were patterned by laser with nine subcells connected in series. The film deposition processes were the same as the normal solar cells as described above. The geometric fill factor (GFF) was calculated to be 90.2%, which was defined as the active area divided by the designated area.

**Characterization**. The GIWAXS was measured with a photon energy of 8.03 keV (1.55 Å) at an incident angle of 0.3°. PL decay kinetics were measured using the Edinburgh Instruments time-correlated single-photon counting (TCSPC) fluorescence spectrometer F900. Voltage-dependent PL decay measurements were performed by varying the reverse bias voltage applied to the cell from 0 to 2.5 V in steps of 0.5 V and simultaneously measuring the PL lifetime at each voltage. The PL decay kinetics were fitted using the intensity-weighted double exponential functions ($\bar{\tau} = \sum_{i=1}^{2} A_i \tau_i^2 / \sum_{i=1}^{2} A_i \tau_i$) to estimate the average lifetimes. The KPFM images were acquired using an MFP-3D Infinity with AC bias modulation at 7.5 KHz and 4 V amplitude in ambient. The UPS was measured by an AXIS Nova spectrometer (Kratos Analytical Ltd, UK) with a He-I source (21.22 eV). The XPS was examined by VersaProbe II (Physical Electronics Inc.) with an Al-Kα source of 1486.6 eV. The micro-PL maps were obtained with an Acton Standard Series SP-2558 laser confocal Raman spectrometer (Princeton Instruments), a digital CCD (PIXIS:100B_eXcelon), and a 485 nm laser (PicoQuant LDH-P-C-485). CL measurements were performed on the Attolight ROSA 4634 CL SEM at 1 kV and 50 nm per pixel for the image of 128 × 128 pixels.

*J*–*V* characteristics of the devices were characterized by a commercial solar simulator (Oriel, 450 W xenon, AAA class) combined with a Keithley 2400 source meter at room temperature in air. The light intensity was calibrated before each measurement with a Si reference cell (KG5, Newport) to match with the AM 1.5 G (100 mW cm$^{-2}$) standard. The voltage scan rate and scan steps were 125 mV s$^{-1}$ and 10 mV, respectively. No light soaking or voltage bias was applied before the measurement and the active area of the cells was defined by a black metal mask with a square aperture of the area of 0.09 cm$^2$. For solar modules, the voltage scan rate was 5 V s$^{-1}$ and scan steps were 50 mV, and the active area is 26 cm$^2$. The statistics of PCEs for devices were obtained from 20 samples for each condition. The EQE spectra were recorded with an IQE200B (Oriel). The stability test was conducted under 100 mW cm$^{-2}$ LED illumination at 25 °C and the devices were sealed in a cell holder flushed with a N$_2$ flow of ~30 mL min$^{-1}$. For the humidity

test, the unsealed solar cells were kept in the ambient of 40–50% RH monitored by a hygrometer at room temperature in the dark. For the thermal stability test, the solar cells were kept at 85 °C in nitrogen under dark conditions. Transient photocurrent measurements were performed with an Agilent Technologies DS05054A oscilloscope using a 1 kΩ input resistor and a Tektronix AFG 3101 function generator. The Mott–Schottky curve was recorded with an SP-200 potentiostat (BioLogic).

**Reporting summary**. Further information on research design is available in the Nature Research Reporting Summary linked to this article.

## Data availability
The data that support the findings of this study are provided in the Supplementary Information/Source Data file. Source data are provided with this paper.

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

## Acknowledgements

S.D. gratefully acknowledges support from the National Key Research and Development Program of China (No. 2020YFB1506400). M.K.N. gratefully acknowledges fundings from the VALAIS ENERGY DEMONSTRATORS FUND and the European Union's Horizon 2020 Research and Innovation programme (No. 763977). Y.D. and S.D. acknowledge sponsorship from the National Natural Science Foundation of China (Nos. U1705256, 51702096, 61904053, 51572080, and 51961165106). S.D. acknowledges financial support from the Higher Education Discipline Innovation Project (No. B16016) and the Fundamental Research Funds for the Central Universities (Nos. 2019MS026 and 2019MS027). Y.Y. and C.L. acknowledge support from the Ph.D. scholarship from China Scholarship Council (Grant Nos. 201906730051 and 201906730050). K.R. acknowledges funding from the Research Council of Lithuania via grant No. S-MIP-20-20 and the funding received from the MJJ Foundation. F.D.A. thanks the Ministero dell'Istruzione dell'Università e della Ricerca (MIUR) and Università degli Studi di Perugia for financial support through the program "Dipartimenti di Eccellenza 2018-2022" (Grant AMIS). E.M. and F.D.A. received funding from PON «R&I» 2014-2020: Progetti di Ricerca Industriale e Sviluppo Sperimentale nelle 12 aree di Specializzazione individuate dal PNR 2015-2020.

## Author contributions

C.L. and Y.Y. contributed equally to this work. C.L., Y.Y., K.R., and M.K.N. conceived the idea. K.R. and R.S. designed and synthesized the organic salts. C.L. and Y.Y. designed the experiments and performed the solar cell and module fabrication, measurement, and optimization. A.M., E.M., and F.D.A. conducted the simulation modeling. M.F., S.D., G.K., and Vi.G. performed the PL decays and transient photocurrent measurement. O.J.U. and J.-N.A. carried out the KPFM measurement. H.K. conducted the XPS and UPS. B.D. and K.G.B. helped with the module fabrication and *J–V* measurements. M.D. performed the CL measurements. K.R., Vy.G., Y.D., S.D., P.J.D., and M.K.N. supervised the project. C.L. and Y.Y. wrote the first draft of the manuscript. All the authors revised and approved the manuscript.

## Competing interests

The authors declare no competing interests.
