## [Peer Review File · Nature Communications]

Tuning structural isomers of phenylenediammonium to afford efficient and stable perovskite solar cells and modulesREVIEWER COMMENTS

Reviewer #1 (Remarks to the Author):

In this manuscript, the authors studied the energy barrier of 2D perovskite formation from ortho-, meta- and para-isomers of (phenylene)di(ethylammonium) iodide (PDEAI2) that were designed for tailored defect passivation. Treatment with the most sterically hindered ortho-isomer not only prevents the formation of surficial 2D perovskite film, even at elevated temperatures, but also maximizes the passivation effect on both shallow- and deep-level defects. The ensuing PSCs achieve an efficiency of 23.9% with long-term operational stability (over 1000 hours). And the authors also demonstrated that a high performance perovskite solar cells module with an active area of 26 cm². Overall, the idea and results shown in this manuscript is good, while still have some issues should be addressed for further considerations:

- 1) For the theoretical modelling, the authors should consider the usually used PEAI for comparison.
- 2) The authors should clearly explain why the passivation effect of the organic salt is better than the 2D perovskite.
- 3) The authors claimed that the record efficiency (21.4%) for the module with the area of 26 cm². The efficiency could be overestimated, for the results shown in Figure 5c, the J_{sc} is 2.71 mA/cm², considering that a nine sub-cells connected, the overall equivalent short current density is 24.39 mA/cm² (this is almost same as single cell current density), while the geometric fill factor (GFF) was calculated to be 90.2%, this means that the real current density will be 27 mA/cm², this is impossible for the current perovskite material. I think the author has deduct the dead area while calculated the current density of the module, this is misleading. If the author insisted on the module efficiency is reliable, the authors should give a certification results from authorized party such as Newport, AIST, NREL or others.

Reviewer #2 (Remarks to the Author):

The authors report a study on organic halide salt passivation to reduce defects in perovskite solar cells (PSCs). The authors studied ortho-, meta- and para-isomers of (phenylene)di(ethylammonium) iodide (PDEAI2) that were used to passivate perovskite defects. They find that the ortho-isomer could prevent the formation of surficial 2D perovskite film for the most sterically hinder, simultaneously maximizing the passivation effect on both shallow- and deep-level defects. At last, the ortho-isomer-based PSCs obtained an efficiency of 23.9%, with high efficiency of 21.4% for the perovskite module (an active area of 26 cm²).

1. Many reports showed that a thin surficial 2D perovskite layer could improve the efficiency of the device, such as the author's work (Sci China Mater 2020, 63(1): 47–54). However, the authors said that the para-isomer that could form surficial 2D perovskite film decreased the device's performance from the PCE of 20.71% (control) to 20.30% (para-isomer).
2. In the assignment of the different phenyl-diethyl ammonium iodide (PDEAI2) solutions post-treatment on the final PSCs performance, only used the concentration of 2 mg mL⁻¹, where there is no optimized process.
3. For surface energy band measurements, the authors said that the downshift suggests a more p-type surface after o-PDEAI2 treatment, which can improve hole extraction and cause band-bending for higher VOC values. However, from the UPS measurements (Supplementary Fig. 7), we could estimate the valence band of perovskite are -5.85 eV for control and ~-6.15 eV for o-PDEAI2 treatment. Thus, this valence band of perovskite seems to impair the hole extraction rather than improve hole extraction as ref 40. (Energy Environ. Sci. 13, 1222-1230 (2020).)
4. In the cathodoluminescence (CL) intensity maps, o-PDEAI2-passivated perovskite film showed low luminescence (CL) intensity at 800 nm, while confocal PL microscopy mapping showed enhanced PL intensity.

5. In the assignment of the passivation effect on the hole extraction and carrier dynamics in the devices, the detailed experiment process of voltage-dependent transient PL measurement should be provided.

Reviewer #3 (Remarks to the Author):

Review "Tuning the structural isomers of phenylenediammonium cations to afford efficient and stable perovskite solar cells and modules"

The manuscript explores approaches to passivate the surface of perovskite materials with 'large' ammonium cations, while at the same time preventing to form a 2D perovskite phases that introduces a substantial barrier for charge extraction. The authors show very convincingly that the use ortho-phenylenediethylammonium is a viable approach to reach this and characterize the different mutual placements of the ethylammonium groups in detail. The study is very thorough and show the effects on the device performance (both in small cells and in modules) very convincingly and is supported by measurements of carrier lifetimes by (voltage dependent) PL and surface potential by Kelvin probe measurements. I believe is an interesting contribution that should attract considerable attention in the perovskite field and hence should be published in Nature Commun. There are a few things that the authors may address to improve the manuscript:

- The structures obtained from the calculations are interesting but it is very hard to make out in the figures provided what they actually look like. For the meta-compound the authors discuss binding of both ammonium groups, does this lead to distortion of the surface, leading to trapping? Does the ortho-compound also bond with both the ammonium groups?
- I believe that some discussion on general design principles that can be derived from this study would be useful. Would the same effects be expected for any ammonium compound that does not have a tendency to form a 2D perovskite phase? What is the relevance of the presence of two alkyl ammoniums? Would a single one do the trick as long as it is attached to an organic group that does not allow 2D-phase formation?

Response Letter to Reviewers' Comments

Response to Reviewer #1:

Comments to the Author: In this manuscript, the authors studied the energy barrier of 2D perovskite formation from ortho-, meta- and para-isomers of (phenylene)di(ethylammonium) iodide (PDEAI₂) that were designed for tailored defect passivation. Treatment with the most sterically hindered ortho-isomer not only prevents the formation of surficial 2D perovskite film, even at elevated temperatures, but also maximizes the passivation effect on both shallow- and deep-level defects. The ensuing PSCs achieve an efficiency of 23.9% with long-term operational stability (over 1000 hours). And the authors also demonstrated that a high performance perovskite solar cells module with an active area of 26 cm². Overall, the idea and results shown in this manuscript is good, while still have some issues should be addressed for further considerations:

Reply: We appreciate the reviewer for his/her valuable suggestions and time to review our manuscript, which improves the quality of the article. According to the reviewer's suggestions, the manuscript has been revised, and its quality has been improved.

(1) For the theoretical modelling, the authors should consider the usually used PEAI for comparison.

Author reply: We thank the reviewer for the valuable comment. According to your suggestion, we have checked the formation energy of PEA₂PbI₄, which results in energetically more feasible (-7.71 eV) compared to *o*-PDEAI₂ (-6.41 eV), *m*-PDEAI₂ (-7.22 eV) and *p*-PDEAI₂ (-7.19 eV). Our simulation results nicely match with experiments. We found that PEAI showed a much stronger ability to react with PbI₂ to form 2D perovskite than the three isomeric cations, as verified by the XRD and PL results (Figure 1d, e and Supplementary Figure 1). We have further checked the passivation effect, following the similar protocol used for *ortho*-, *meta*- and *para*-isomers, of PEA cation by adsorbing on perovskite surface (see **Figure R1**). The adsorption energy results in -9.28 and -9.90 eV for the ethyl ammonium and phenyl ring adsorption modes, respectively. The adsorption energy values are similar to the *o*-PDEAI₂, which are -10.00 and -9.53 eV for the ethyl ammonium and phenyl ring adsorption modes, respectively. Thus, the adsorption energy suggests that PEAI has a similar passivation effect with *o*-PDEAI₂, consistent with previous reports.¹

The corresponding description and figure of the theoretical calculation for PEAI have been added to the revised manuscript and supplementary information.

Figure R1. Optimized structures for the adsorption of PEA cation on the perovskite surface; (a) through -NH_3 and (b) through phenyl group.

(2) The authors should clearly explain why the passivation effect of the organic salt is better than the 2D perovskite.

Author reply: We would like to thank the reviewer for the precious question. The surface layers of 2D perovskite and organic halide salts can provide a chemical passivation effect for solution-processed 3D perovskite films on surficial defects, which contribute to interface carrier recombination and limit device performance well below their Shockley-Queisser efficiency potential.²

For 2D perovskite passivation, organic spacer cations are used to react with excessive PbI_2 or to be exchanged with formamidinium (FA) and methylammonium (MA) cations to form a 2D perovskite capping layer, where a post-deposition thermal annealing step is usually involved to convert the film to the 2D phase. After covering the 3D perovskite, the 2D perovskite layer with upwardly shifting band edges may facilitate the hole extraction from the bottom 3D perovskite to hole transport layer (HTL) and suppresses the interfacial electron-hole recombination, reducing the energy loss inside 2D/3D perovskite solar cell (PSCs). However, several issues still challenge this strategy. (1) The 2D perovskite exhibits a high optical bandgap (≈ 2.4 eV) leading to a shorter spectral window for light absorption (300-500 nm) and typically has a much higher exciton binding energy (300-500 meV) than 3D perovskite, which hinders charge separation.^{3,4} (2) The stubborn in-plane orientation of 2D perovskite impedes vertical hole transport in devices. Specifically, the organic spacers form thin insulating layers between the more conductive metal halide slabs, adversely affecting the conductivity of 2D perovskite films and impeding charge extraction, as shown in **Figure R2**.⁵ This requires ultrathin 2D perovskite layers or incomplete 2D perovskite

surface coverage to permit charge tunnelling or extraction via unpassivated regions.^{1,6-8} Therefore, optimizing charge transfer in 2D/3D PSCs for further improvement in device performance remains a challenge.

We have made some attempts to improve the performance of 2D/3D PSCs. During our exploration, it was found that the efficiencies of PSCs passivated with organic halide salts, such as the widely used phenyl ethyl ammonium iodide (PEAI), without post-treatment thermal annealing are usually higher than that of with thermal annealing (**Figure R3**). And the difference in device performance derived from whether the 2D perovskite was formed or not. Similar results were also reported by You et.al.¹ Therefore, removing the post-treatment annealing step to enable an organic halide salt capping layer is considered as a more promising passivation strategy, which has been verified by the state-of-the-art PSCs have verified with efficiencies of around 25% published in *Nature* and *Science*.⁹⁻¹¹ Organic halide salts reported so far for surface passivation mainly include alkylammonium halides and phenyl ammonium halides. This type of material always has a comprehensive passivation effect on both the cation and anion defects, and enhances the adhesion to the 3D perovskite film through hydrogen bonding or ionic bonding. The ammonium group is thought to form a hydrogen bond with the iodide ion on the 3D perovskite surface;¹² The π -conjugation structure of the aromatic rings can capture electronic defects pertaining to trace amount of neutral iodine;¹³ The iodide ions can fill the iodine vacancies and coordinate with the Pb^{2+} interstitials.¹² Therefore, organic halide salts can provide an effective passivation effect on the perovskite surface, and avoid the high exciton binding energy and adverse in-plane orientation of 2D perovskite.

Even so, PSCs passivated with the commonly used PEA salt suffer from the efficiency plummet under higher operating temperatures due to the inevitable conversion to 2D perovskite.¹ In fact, we also found that this conversion could happen even under AM 1.5G light illumination, which would weaken the passivation effect of organic halide salt. Therefore, we developed an *ortho*-(phenylene)di(ethyl ammonium) iodide (*o*-PDEAI₂) that can sustain higher temperatures without undergoing 2D perovskite formation to effectively passivate surface defects via dual -NH₃I terminals, the phenyl rings, and the iodide ions. Enhancing the formation energy of 2D perovskite by tuning the spatial structure of bulky organic cations is the key factor to prevent them from entering the perovskite lattice and forming 2D perovskite even at elevated temperatures. In comparison, post-treatment with *para*-(phenylene)di(ethyl ammonium) iodide (*p*-PDEAI₂) easily induced the formation of 2D perovskite due to the lower energy barrier and led to lower device performance than that of *o*-PDEAI₂ because of the problems mentioned above of 2D/3D structure, such as the higher binding energy and the hindered charge transport by the unfavorable in-plane orientation. Although PEA and PDEAI₂ passivation exhibits a better passivation effect than their 2D perovskites, the relative lack of studies that directly compare the performance of devices passivated with pristine organic halide salts and their 2D perovskites makes it difficult to assess the generality of the above observations. The type, location, quantity, or other factors of functional groups on organic ammonium

cations should influence the passivation properties of organic halide salts and their 2D perovskites, which deserves further systematic studies in the future.

We have added a brief description regarding the reason for the improved passivation effect of organic salts to the introduction and photovoltaic performance sections in the revised manuscript.

Figure R2. Schematic illustration of 2D perovskite and organic halide salt-assembled perovskite surface.

Figure R3. *J-V* characteristics of the PEAI-passivated devices with and without thermal annealing.

(3) The authors claimed that the record efficiency (21.4%) for the module with the area of 26 cm². The efficiency could be overestimated, for the results shown in Figure 5c, the J_{sc} is 2.71 mA/cm², considering that a nine sub-cells connected, the overall equivalent short current density is 24.39 mA/cm² (this is almost same as single cell current density), while the geometric fill factor (GFF) was calculated to be 90.2%, this means that the real current density will be 27 mA/cm², this is impossible for the current perovskite material. I think the author has deduct the dead area while calculated the current density of the module, this is misleading. If the author insisted on the module efficiency is reliable, the

authors should give a certification results from authorized party such as Newport, AIST, NREL or others.

Author reply: Thank you for your precious question. Sorry for the confusion. As we described in the manuscript, 26 cm^2 is the active area, obtained by deducting the dead area from the designated area. And the performance parameters of the modules including J_{SC} of 2.71 mA/cm^2 were calculated based on the active area instead of the designated area. Thus, the real J_{SC} was 24.39 mA/cm^2 , which was slightly lower than that of the small cell (24.75 mA/cm^2) and in the reasonable range of J_{SC} values for the FA-based PSCs. Based on the designated area, the J_{SC} and PCE were calculated to be 2.44 mA/cm^2 (overall equivalent J_{SC} of 22.00 mA/cm^2) and 19.27%, respectively.

The reason why we mainly used the active area to calculate the performance parameters is as follows: The GFF is highly dependent on how advanced the lab-based laser etching technology, and there can be significant differences among different research groups who are mostly focusing on the basic research. Therefore, the module performance parameters based on the active area is also very useful to compare the effectiveness of materials and experimental methods on large-scale application among different research groups, as reported by many recent publications, such as [*Science* **372**, 1327-1332 (2021); *Nature Energy* **6**, 633-641 (2021); *Joule* **5**, 958-974 (2021); *Joule* **4**, 2675-2692 (2020); *Energy & Environmental Science* (2021) DOI: 10.1039/D1EE01440D; *Advanced Materials* **32**, 2004979 (2020); *Joule* **4**, 2675-2692 (2020)].

Indeed, we understand that the evaluation of module performance using the designated area is necessary to estimate the real performance in practical application. Therefore, we have added the designated-area performance of perovskite solar modules to the revised manuscript. Also, we removed the text “ 26 cm^2 ” from Figure 5a to avoid possible misunderstanding.

Response to Reviewer #2:

Comments to the Author: The authors report a study on organic halide salt passivation to reduce defects in perovskite solar cells (PSCs). The authors studied ortho-, meta- and para-isomers of (phenylene)di(ethylammonium) iodide (PDEAI₂) that were used to passivate perovskite defects. They find that the ortho-isomer could prevent the formation of surficial 2D perovskite film for the most sterically hinder, simultaneously maximizing the passivation effect on both shallow- and deep-level defects. At last, the ortho-isomer-based PSCs obtained an efficiency of 23.9%, with high efficiency of 21.4% for the perovskite module (an active area of 26 cm^2).

Reply: We thank the reviewer for the time spent reviewing our work. We appreciated the constructive points raised, which we believed helped in improving the quality of our work. Here below, the specific comments have been considered, and action has been taken to fulfil the reviewer request.

(1) Many reports showed that a thin surficial 2D perovskite layer could improve the efficiency of the device, such as the author's work (Sci China Mater 2020, 63(1): 47-54). However, the authors said that the para-isomer that could form surficial 2D perovskite film decreased the device's performance from the PCE of 20.71% (control) to 20.30% (para-isomer).

Author reply: We appreciate the reviewer for the careful review and precious question. As the reviewer mentioned, some 2D perovskites have been reported to have a passivation effect and improve the device performance. We also have been focusing on this topic and screened out some suitable 2D perovskite passivators in the past few years.¹⁴⁻¹⁹ The 2D perovskite capping layer with upwardly shifting band edges could facilitate the hole extraction and block the charge recombination at the perovskite/HTL interface to improve the device performance. However, there are also two intrinsic issues of 2D perovskite adverse to the device performance. (1) 2D perovskite exhibits inferior optoelectrical properties for photovoltaic application. The high optical bandgap (≈ 2.4 eV) shortens the spectral window for light absorption. The high exciton binding energy (300-500 meV) hinders charge separation. The poor electrical conductivity (10^{-5} S m^{-1}) inhibit the charge transport.^{3,4,20} (2) The in-plane orientation of surficial 2D perovskite films was commonly observed due to the minimized surface energy, while this hinders the out-of-plane charge transport and leads to a high degree of electrical anisotropy because of the insulating character of the bulky organic cations.⁵ Therefore, the influence of 2D perovskite passivation on device performance depends on the comprehensive effect of these above-mentioned factors. It cannot be concluded that all 2D perovskite materials can behave as good passivators. The type, location, quantity, and other factors of functional groups on organic ammonium cations directly influence the 2D perovskite properties and thus their passivation effect. For example, some 2D perovskites showed adverse passivation effect and decreased device performance, such as those based on 1,4-butanediammonium iodide (C4), 2-[2-(2-aminoethoxy)ethoxy]ethan-1-ammonium iodide (EDBE), and *p*-phenyl diammonium iodide (PDAI) as reported before and also *para*-(phenylene)di(ethylammonium) iodide (*p*-PDEAI₂) in this work.^{13,21} A general rule to judge which type of 2D perovskites can provide an overall positive passivation effect for enhancing the device performance is worth to be systematically investigated and established in the future.

The explanation for the slight decrease in efficiency for *p*-PDEAI₂-based devices has been added to the revised manuscript.

(2) In the assignment of the different phenyl-diethyl ammonium iodide (PDEAI₂) solutions post-treatment on the final PSCs performance, only used the concentration of 2 mg mL⁻¹, where there is no optimized process.

Author reply: We thank the reviewer for the helpful advice. The optimization of the PDEAI₂ concentrations has been conducted as shown in **Table R1**. The highest efficiency was achieved by *o*-PDEAI₂ passivation with the concentration of 2 mg mL⁻¹. For *p*-PDEAI₂ passivation, over 21% efficiencies were achieved by 1 and 2 mg

mL^{-1} , while further increasing the concentration would decrease the efficiency. For *m*-PDEAI₂ passivation, the efficiencies continuously decreased as the concentration increased. The detailed optimization of concentrations of PDEAI₂ post-treatment has been added to the Supplementary Information as Supplementary Table 4.

Table R1. Summary of the device performance with different PDEAI₂ concentrations.

Condition	Concentration (mg mL ⁻¹)	V _{OC} (V)	J _{sc} (mA cm ⁻²)	FF	PCE (%)
Control	0	1.135	24.49	0.790	21.94
	1	1.129	24.31	0.766	21.05
p -PDEAI ₂	2	1.127	24.34	0.770	21.09
	3	1.131	23.90	0.767	20.70
	4	1.117	23.87	0.765	20.43
	5	1.115	23.83	0.751	19.94
	1	1.098	23.77	0.762	19.87
m -PDEAI ₂	2	1.079	23.55	0.750	19.63
	3	1.063	24.14	0.720	18.48
	4	1.061	23.95	0.703	17.91
	5	1.048	22.04	0.708	16.32
	1	1.147	24.64	0.822	23.23
o -PDEAI ₂	2	1.157	24.75	0.835	23.92
	3	1.153	24.60	0.832	23.63
	4	1.151	24.46	0.816	22.98
	5	1.148	24.37	0.799	22.36

(3) For surface energy band measurements, the authors said that the downshift suggests a more p-type surface after *o*-PDEAI₂ treatment, which can improve hole extraction and cause band-bending for higher V_{OC} values. However, from the UPS measurements (Supplementary Fig. 7), we could estimate the valence band of perovskite are -5.85 eV for control and ~6.15 eV for *o*-PDEAI₂ treatment. Thus, this valence band of perovskite seems to impair the hole extraction rather than improve hole extraction as ref 40. (Energy Environ. Sci. 13, 1222-1230 (2020).)

Author reply: Thank you for your valuable question. Instead of 2D perovskite passivation, which introduced a new large-bandgap perovskite material between the 3D perovskite and the hole transport material (HTM), in this work, *o*-PDEAI₂ passivation adjusted the electronic band structures while maintained the bandgap of the perovskite film (**Figure R4**). The UPS results revealed downshifted Fermi level (E_F) and valence band (E_V), indicating a less n-type surface after *o*-PDEAI₂ passivation. The downshifted E_F and E_V of passivated perovskite have been demonstrated to promote hole extraction by many passivators, such as theophylline, choline iodine, azetidinium, perhydropoly(silazane), 1-(4-bromophenyl)-6,7-diphenylimidazo[5,1,2-cd]indolizine, benzopentafulvalenes

compound.²²⁻²⁷ In fact, the energy level distribution for the passivated perovskite film and the formation reason are complicated. As shown in **Figure R5**, since the more n-type region has a higher electron concentration than the less n-type region, electrons diffuse from the more n-type side to the less n-type side. When the electrons move away, they leave exposed positive ion cores in a more n-type side because they are fixed in the crystal lattice and cannot move. Similarly, on the less n-type side, negative ion cores are exposed. An electric field forms between the two regions, which is responsible for the drift of electrons from the less to more n-type side and hinders the diffusion of electrons. During this process, the E_F along with the energy level of the more n-type region moves down, while E_F along with the energy level of the less n-type region moves up until an equilibrium situation of diffusion and drift is reached and the two E_F is levelled at the same time. This causes a bending of the energy bands with the upwards band bending moving from more n-type to less n-type region. The same bandgap of perovskite in both regions determines that the E_V of the more n-type side is lower than that of the less n-type side after bending, which indicates an effective energy level alignment to promote hole extraction and transport.

In fact, taking advantage of the energy band bending to improve the hole transport and extraction has also been reported in 2D/3D PSCs.^{28,29} However, this strategy is not feasible for all cases of 2D/3D structure because the energy band bending is influenced by the E_F , E_V and bandgap of the 2D perovskite. For example, the 2-thiophenemethylammonium chloride (2-TMACl)-based 2D perovskite showed an improper energy band bending with 3D perovskite because of its excessively high E_F and low E_V values.²⁸ This is possibly one of the reasons why different 2D perovskites have different passivation effects.

Figure R4. Tauc plot of the perovskite films without and with *o*-PDEAl₂ passivation.

Figure R5. Illustration of proposed energy band bending of the perovskite film with *o*-PDEAl₂ passivation.

(4) In the cathodoluminescence (CL) intensity maps, *o*-PDEAl₂-passivated perovskite film showed low luminescence (CL) intensity at 800 nm, while confocal PL microscopy mapping showed enhanced PL intensity.

Author reply: Thank you for your careful review. Confocal PL microscopy mapping, which is conducted using a laser to stimulate the light emission of samples after the absorption of photons, is an optical measurement. PL intensity is only determined by the power and wavelength of the excitation light and sample properties. Thus, the PL mapping intensity of different samples can be compared if applying the same laser. Different from the PL measurement, CL is an optical and electromagnetic measurement in which light emission of samples is obtained by excitation of a high-energy electron beam, typically produced in an electron microscope, such as SEM, TEM, or microprobe EPMA. In addition to the sample properties, CL intensity largely depends on the measurement conditions. However, each sample usually has different measurement conditions, such as focus, depth of field, beam accelerating voltage, probe current, spot size, and probe convergence angle, which are interlocking and very sensitive to the sample conductivity, working distance, contamination, and artificial factor. Therefore, CL intensities are hard to be compared among different samples due to their variable beam intensities caused by different measurement conditions. While the beam condition is exactly the same within one sample, so the CL intensity at different positions can be compared to judge the uniformity of the luminescence.

(5) In the assignment of the passivation effect on the hole extraction and carrier dynamics in the devices, the detailed experiment process of voltage-dependent transient PL measurement should be provided.

Author reply: We thank the reviewer for the comment. PL decay kinetics were measured using the Edinburgh Instruments time-correlated single-photon counting (TCSPC) fluorescence spectrometer F900. The picosecond pulsed diode laser EPL-470 emitting 72 ps pulses at 470 nm with the repetition rate of 500 kHz (interval 2 μ s) was used for the sample excitation. The time resolution of the setup was about several hundreds of picoseconds by applying apparatus function deconvolution. Voltage-dependent PL decay measurements were performed by varying the reverse bias voltage applied to the cell from 0 to 2.5 V in steps of 0.5 V and simultaneously measuring the PL lifetime at each voltage. The PL decay kinetics were fitted using the intensity-weighted double exponential functions ($\bar{\tau} = \sum_{i=1}^2 A_i \tau_i^2 / \sum_{i=1}^2 A_i \tau_i$) to estimate the average lifetimes.

The primary experiment process of voltage-dependent transient PL measurement has been provided in the characterization of the revised manuscript.

Response to Reviewer #3:

Comments to the Author: The manuscript explores approaches to passivate the surface of perovskite materials with ‘large’ ammonium cations, while at the same time preventing to form a 2D perovskite phases that introduces a substantial barrier for charge extraction. The authors show very convincingly that the use ortho-phenylenediethylammonium is a viable approach to reach this and characterize the different mutual placements of the ethylammonium groups in detail. The study is very thorough and show the effects on the device performance (both in small cells and in modules) very convincingly and is supported by measurements of carrier lifetimes by (voltage dependent) PL and surface potential by Kelvin probe measurements. I believe is an interesting contribution that should attract considerable attention in the perovskite field and hence should be published in Nature Commun. There are a few things that the authors may address to improve the manuscript:

Reply: We thank the reviewer for the time spent reviewing our work. We appreciated the constructive comments and suggestions, which we believed helped improve the quality of our work. According to the reviewer’s suggestions, the manuscript has been revised and its quality has been improved.

(1) The structures obtained from the calculations are interesting but it is very hard to make out in the figures provided what they actually look like. For the meta-compound the authors discuss binding of both ammonium groups, does this lead to distortion of the surface, leading to trapping? Does the ortho-compound also bond with both the ammonium groups?

Author reply: Thank you for your valuable question. To facilitate reading, we have provided the supporting figures with the zoomed view of the different structures focusing on the adsorbed molecule, see **Figure R6**. The adsorption of all the three isomeric cations leads to some surface reconstruction due to its favourable binding tendency. However, in this work, we have not looked into the aspects of the formation

possibility of surface defect traps and we feel that it is beyond the scope of this study. Nevertheless, the increased binding affinity of the ortho cation is associated with the reduced defect formation possibility, which is likely the reason for the improved defect passivation in the ortho isomer. In all three isomeric cations, we attempted to check the adsorption behaviours through both the two ammonium groups and phenyl rings. Upon relaxation, all the three isomers bind with both the ammonium groups. On the other hand, while binding through the phenyl ring, the cations orient themselves to some extent in order to get the local minima structures. The supporting zoomed view of the different adsorption modes is shown in **Figure R6**.

The corresponding description and zoomed figure have been added to the revised manuscript and supplementary information.

Figure R6. Optimized structures for the adsorption of (a-b) ortho-, (c-d) meta- and (e-f) para cation on the perovskite surface. Left and right panel represents the adsorption through ammonium and phenyl group, respectively.

(2) I believe that some discussion on general design principles that can be derived from this study would be useful. Would the same effects be expected for any ammonium compound that does not have a tendency to form a 2D perovskite phase? What is the relevance of the presence of two alkyl ammoniums? Would a

single one do the trick as long as it is attached to an organic group that does not allow 2D-phase formation?

Author reply: We thank the reviewer for the precious suggestion. This work also attempts to explore the design principles for organic halide salt passivation of perovskites. Based on our experience and knowledge, we believe that the influence of organic halide salts on the passivation effect of perovskite is complicated. (1) Not all organic halide salts, which do not form 2D perovskites have a positive passivation effect. For example, besides *meta*-(phenylene)di(ethyl ammonium) iodide (*m*-PDEAI₂) in this work, we also found that passivation with hexa-2,4-diyne-1,6-diammonium iodide (HDDAI) decreased the device performance probably due to its overhigh electron-withdrawing capability. (2) Two alkyl ammoniums normally endow the organic cation with stronger adsorption on the perovskite surface. For example, the adsorption energies of *o*-PDEA (-10.00 eV), *m*-PDEA (-10.50 eV) and *p*-PDEA (-9.76 eV) cations are higher than that of the PEA cation (-9.28 eV) for the ethyl ammonium adsorption modes (Supplementary Table 2). This higher adsorption energy usually enables more organic cations to adhered to the perovskite surface for better defect passivation. However, this is not absolute, because the passivation effect is a result of comprehensive factors. Some diammonium cations decreasing the device efficiency after surface passivation, such as *m*-PDEAI₂ and *p*-phenyl diammonium iodide (PDAI).¹³ (3) Single ammonium organic halide salts including phenyl ethyl ammonium iodide (PEAI), octyl ammonium iodide (OAI), dodecyl ammonium iodide (DAI), 4-*tert*-butyl-benzyl ammonium iodide (tBBAI), and 1,4-octane diammonium iodide (C8), have been demonstrated to have superior passivation effect, while this was achieved by removing the post-treatment thermal annealing to suppress the formation of 2D perovskite.^{1,11,21,30} For the single ammonium organic cation, which inherently inhibits the formation of 2D perovskite, *tert*-butylammonium (tBA) was reported to passivate interface defects and improve device performance. Whereas and the reason for the suppression of the 2D phase is that tBA is a small size cation, which can be incorporated into the 3D perovskite lattice by ion exchange and reaction with PbI₂.³¹ Although there is still a lack of reported cases, we believe that bulky single ammonium organic cations, which do not allow 2D perovskite formation by altering organic groups, should provide good passivation effect if the functional groups are also favorable.

Overall, we speculate that: (1) The type, structure, location, quantity, or other factors of functional groups on ammonium cations comprehensively impact the passivation properties of organic halide salts. The widely used functional groups in passivating organic cations are alkyl aromatic groups and long alkyl chains. In consideration of the good passivation effects of N, F, S, and O heteroatoms and their related functional groups in additive engineering, they can also be incorporated into organic halide salts to further enhance the passivation effect in the future. (2) Preventing the organic cations from forming 2D perovskites under thermal and light conditions is crucial to ensure their continuous and stable passivation effect. Based on our experience, regulation of the molecule structure to adjust the chemical

environment and steric effect of ammoniums may effectively increase the formation energy barrier of 2D perovskites, as revealed by this work.

A brief outlook has been added to the conclusion in the revised manuscript.

Figure R7. Molecule Structure of the organic halide salts.

References

- 1 Jiang, Q. *et al.* Surface passivation of perovskite film for efficient solar cells. *Nature Photonics* **13**, 460-466, doi:10.1038/s41566-019-0398-2 (2019).
- 2 Mahmud, M. A. *et al.* Origin of Efficiency and Stability Enhancement in High-Performing Mixed Dimensional 2D-3D Perovskite Solar Cells: A Review. *Advanced Functional Materials*, 2009164 (2021).
- 3 Lin, D. *et al.* Stable and scalable 3D-2D planar heterojunction perovskite solar cells via vapor deposition. *Nano Energy* **59**, 619-625, doi:https://doi.org/10.1016/j.nanoen.2019.03.014 (2019).
- 4 Huang, W., Sadhu, S., Sapkota, P. & Ptasinska, S. In situ identification of cation-exchange-induced reversible transformations of 3D and 2D perovskites. *Chemical Communications* **54**, 5879-5882, doi:10.1039/C8CC02801J (2018).
- 5 Yang, Y. *et al.* Universal approach toward high-efficiency two-dimensional perovskite solar cells via a vertical-rotation process. *Energy & Environmental Science* **13**, 3093-3101, doi:10.1039/D0EE01833C (2020).
- 6 Yoo, J. J. *et al.* An interface stabilized perovskite solar cell with high stabilized efficiency and low voltage loss. *Energy & Environmental Science* **12**, 2192-2199, doi:10.1039/C9EE00751B (2019).
- 7 Mahmud, M. A. *et al.* Double-Sided Surface Passivation of 3D Perovskite Film for High-Efficiency Mixed-Dimensional Perovskite Solar Cells. *Advanced Functional Materials* **30**, 1907962, doi:https://doi.org/10.1002/adfm.201907962 (2020).

- 8 Duong, T. *et al.* High Efficiency Perovskite-Silicon Tandem Solar Cells: Effect of Surface Coating versus Bulk Incorporation of 2D Perovskite. *Advanced Energy Materials* **10**, 1903553, doi:<https://doi.org/10.1002/aenm.201903553> (2020).
- 9 Jeong, J. *et al.* Pseudo-halide anion engineering for α -FAPbI₃ perovskite solar cells. *Nature* **592**, 381-385, doi:10.1038/s41586-021-03406-5 (2021).
- 10 Kim, G. *et al.* Impact of strain relaxation on performance of α -formamidinium lead iodide perovskite solar cells. *Science* **370**, 108-112, doi:10.1126/science.abc4417 (2020).
- 11 Min, H. *et al.* Efficient, stable solar cells by using inherent bandgap of α -phase formamidinium lead iodide. *Science* **366**, 749-753, doi:10.1126/science.aay7044 (2019).
- 12 Gao, F., Zhao, Y., Zhang, X. & You, J. Recent progresses on defect passivation toward efficient perovskite solar cells. *Advanced Energy Materials* **10**, 1902650 (2020).
- 13 Hou, M. *et al.* Aryl Diammonium Iodide Passivation for Efficient and Stable Hybrid Organ-Inorganic Perovskite Solar Cells. *Advanced Functional Materials* **30**, 2002366, doi:<https://doi.org/10.1002/adfm.202002366> (2020).
- 14 Grancini, G. *et al.* One-Year stable perovskite solar cells by 2D/3D interface engineering. *Nature communications* **8**, 1-8 (2017).
- 15 Cho, K. T. *et al.* Selective growth of layered perovskites for stable and efficient photovoltaics. *Energy & Environmental Science* **11**, 952-959 (2018).
- 16 Cho, K. T. *et al.* Water-Repellent Low-Dimensional Fluorous Perovskite as Interfacial Coating for 20% Efficient Solar Cells. *Nano Letters* **18**, 5467-5474, doi:10.1021/acs.nanolett.8b01863 (2018).
- 17 Hu, R. *et al.* Enhanced stability of α -phase FAPbI₃ perovskite solar cells by insertion of 2D (PEA)₂PbI₄ nanosheets. *Journal of Materials Chemistry A* **8**, 8058-8064, doi:10.1039/C9TA14207J (2020).
- 18 Kim, H. *et al.* Self-Crystallized Multifunctional 2D Perovskite for Efficient and Stable Perovskite Solar Cells. *Advanced Functional Materials* **30**, 1910620, doi:<https://doi.org/10.1002/adfm.201910620> (2020).
- 19 Paek, S. *et al.* Molecular Design and Operational Stability: Toward Stable 3D/2D Perovskite Interlayers. *Advanced Science* **7**, 2001014, doi:<https://doi.org/10.1002/advs.202001014> (2020).
- 20 Ortiz-Cervantes, C., Román-Román, P. I., Vazquez-Chavez, J., Hernández-Rodríguez, M. & Solis-Ibarra, D. Thousand-fold Conductivity Increase in 2D Perovskites by Polydiacetylene Incorporation and Doping. *Angewandte Chemie International Edition* **57**, 13882-13886, doi:<https://doi.org/10.1002/anie.201809028> (2018).
- 21 Zhao, T., Chueh, C.-C., Chen, Q., Rajagopal, A. & Jen, A. K. Y. Defect Passivation of Organic-Inorganic Hybrid Perovskites by Diammonium Iodide toward High-Performance Photovoltaic Devices. *ACS Energy Letters* **1**, 757-763, doi:10.1021/acsenerylett.6b00327 (2016).
- 22 Wang, R. *et al.* Constructive molecular configurations for surface-defect

- passivation of perovskite photovoltaics. *Science* **366**, 1509-1513 (2019).
- 23 Wang, Y. *et al.* Thermodynamically stabilized β -CsPbI₃-based perovskite solar cells with efficiencies >18%. *Science* **365**, 591-595, doi:10.1126/science.aav8680 (2019).
- 24 Ansari, F. *et al.* Passivation Mechanism Exploiting Surface Dipoles Affords High-Performance Perovskite Solar Cells. *Journal of the American Chemical Society* **142**, 11428-11433, doi:10.1021/jacs.0c01704 (2020).
- 25 Kanda, H. *et al.* Band-bending induced passivation: high performance and stable perovskite solar cells using a perhydropoly (silazane) precursor. *Energy & Environmental Science* **13**, 1222-1230 (2020).
- 26 Wen, L. *et al.* Reducing Defects Density and Enhancing Hole Extraction for Efficient Perovskite Solar Cells Enabled by π -Pb²⁺ Interactions. *Angewandte Chemie International Edition* **60**, 17356-17361, doi:https://doi.org/10.1002/anie.202102096 (2021).
- 27 Guo, X. *et al.* Synergetic surface charge transfer doping and passivation toward high efficient and stable perovskite solar cells. *iScience* **24**, 102276, doi:https://doi.org/10.1016/j.isci.2021.102276 (2021).
- 28 Sutanto, A. A. *et al.* 2D/3D perovskite engineering eliminates interfacial recombination losses in hybrid perovskite solar cells. *Chem* **7**, 1903-1916, doi:10.1016/j.chempr.2021.04.002 (2021).
- 29 Huang, P.-C. *et al.* Visualizing band alignment across 2D/3D perovskite heterointerfaces of solar cells with light-modulated scanning tunneling microscopy. *Nano Energy* **89**, 106362, doi:https://doi.org/10.1016/j.nanoen.2021.106362 (2021).
- 30 Zhu, H. *et al.* Tailored Amphiphilic Molecular Mitigators for Stable Perovskite Solar Cells with 23.5% Efficiency. *Advanced Materials* **32**, 1907757, doi:https://doi.org/10.1002/adma.201907757 (2020).
- 31 Bu, T. *et al.* Surface modification via self-assembling large cations for improved performance and modulated hysteresis of perovskite solar cells. *Journal of Materials Chemistry A* **7**, 6793-6800, doi:10.1039/C8TA12284A (2019).

REVIEWERS' COMMENTS

Reviewer #1 (Remarks to the Author):

The authors have addressed the Reviewer's concerns and also revised the manuscript accordingly. Now the paper is ready for publication.

Reviewer #2 (Remarks to the Author):

The comments were answered properly. The work towards module fabrication is meaningful to the development of the perovskite solar cells.

Reviewer #3 (Remarks to the Author):

The authors have made significant clarifications to the manuscript in response to my earlier comments. I believe the manuscript can be published in this form.

Response Letter to Reviewers' Comments

Response to Reviewer #1:

Comments to the Author: The authors have addressed the Reviewer's concerns and also revised the manuscript accordingly. Now the paper is ready for publication.

Reply: We thank the reviewer for the recommendation.

Response to Reviewer #2:

Comments to the Author: The comments were answered properly. The work towards module fabrication is meaningful to the development of the perovskite solar cells.

Reply: We thank the reviewer for the high appraisal and the recommendation.

Response to Reviewer #3:

Comments to the Author: The authors have made significant clarifications to the manuscript in response to my earlier comments. I believe the manuscript can be published in this form.

Reply: We thank the reviewer for the recommendation.